# REWARD DESIGN WITH LANGUAGE MODELS

**Minae Kwon, Sang Michael Xie, Kalesha Bullard[†], Dorsa Sadigh**
Stanford University, DeepMind[†]
{minae, xie, dorsa}@cs.stanford.edu, ksbullard@deepmind.com[†]

## ABSTRACT

Reward design in reinforcement learning (RL) is challenging since specifying human notions of desired behavior may be difficult via reward functions or require many expert demonstrations. Can we instead cheaply design rewards using a natural language interface? This paper explores how to simplify reward design by prompting a large language model (LLM) such as GPT-3 as a proxy reward function, where the user provides a textual prompt containing a *few examples* (few-shot) or a *description* (zero-shot) of the desired behavior. Our approach leverages this proxy reward function in an RL framework. Specifically, users specify a prompt once at the beginning of training. During training, the LLM evaluates an RL agent's behavior against the desired behavior described by the prompt and outputs a corresponding reward signal. The RL agent then uses this reward to update its behavior. We evaluate whether our approach can train agents aligned with user objectives in the Ultimatum Game, matrix games, and the DEALORNODEAL negotiation task. In all three tasks, we show that RL agents trained with our framework are well-aligned with the user's objectives and outperform RL agents trained with reward functions learned via supervised learning. Code and prompts can be found here.

## 1 INTRODUCTION

Autonomous agents are becoming increasingly capable with the rise of compute and data. This underscores the importance for human users to be able to control what policies the agents learn and ensure the policies are aligned with their objectives. For instance, imagine training an agent to represent users in a salary negotiation. A working mother fighting for a livable wage may want their agent to be stubborn whereas a new hire looking to develop a good relationship with the company may want their agent to be more versatile.

Currently, users specify desired behaviors by 1) designing reward functions or 2) providing large amounts of labeled data. Both approaches are challenging and impractical for different reasons. Designing reward functions is not an intuitive way to specify preferences. For instance, it isn't straightforward how to write a reward function for a "versatile" negotiator. Furthermore, designing reward functions that balance between different objectives — also known as the "reward design problem" — is notoriously difficult because agents are susceptible to reward hacking (Amodei et al., 2016; Hadfield-Menell et al., 2017). On the other hand, one can learn a reward function from labeled examples. However, that is not possible with a single example; we need large amounts of labeled data to capture the nuances of different users' preferences and objectives, which has shown to be costly (Zhang et al., 2016). Additionally, both approaches do not generalize well to new users who have different objectives — we would have to re-design our reward functions or re-collect data.

Our aim is to create an easier way for users to communicate their preferences, where the interface is more intuitive than crafting a reward function and where they can cheaply specify their preferences with no more than a few examples. To do this, we leverage large language models (LLMs) that are trained on internet-scale text data and have shown an impressive ability to learn in-context from few or zero examples (Brown et al., 2020). Our key insight is that

> The scale of data that LLMs have been trained on make them great in-context learners and also allows them to capture meaningful commonsense priors about human behavior. Given a few examples or a description demonstrating the user's objective, an LLM should be able to provide an accurate instantiation of reward values on a new test example, allowing for easier generalization to new objectives.

To this end, we explore how to prompt an LLM as a proxy reward function to train RL agents from user inputs. In our approach, the user specifies an objective with a natural language prompt. Objectives can

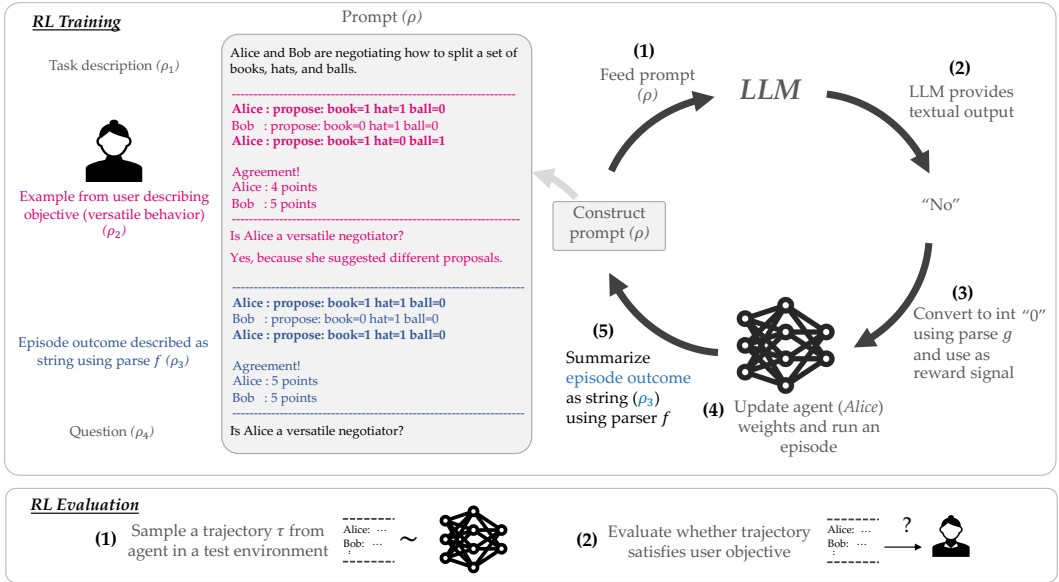

Figure 1: Depiction of our framework on the DEALORNODEAL negotiation task. A user provides an example and explanation of desired negotiating behavior (e.g., versatility) before training. During training, (1) we provide the LLM with a task description, a user's description of their objective, an outcome of an episode that is converted to a string, and a question asking if the outcome episode satisfies the user objective. (2-3) We then parse the LLM's response back into a string and use that as the reward signal for the *Alice* the RL agent. (4) *Alice* updates their weights and rolls out a new episode. (5) We parse the episode outcome int a string and continue training. During evaluation, we sample a trajectory from *Alice* and evaluate whether it is aligned with the user's objective.

be specified with a few examples when they are difficult to define (such as "versatility") or as a single phrase when they are well-known concepts (such as "Pareto-optimality"). We use the prompt and the LLM to define a reward function for training an RL agent. The LLM takes the user prompt and a trajectory from an RL episode as input and outputs a score (e.g., "No" or "0") for whether the trajectory satisfies the user's objective, which we parse as an integer reward for the RL agent (Figure 1).

There are two advantages to prompting LLMs as a proxy reward function: (1) we can leverage LLM's in-context learning abilities and prior knowledge on human behavior so that users only need to provide a handful of example desirable behaviors and (2) users can specify their preferences intuitively using language. On the other hand, a potential disadvantage is that it is unclear how much prompt design will be required for the LLM to reliably infer user intent (see Sec. 5 for a discussion). The goal of this paper is to explore how well LLMs can train objective-aligned agents by providing reward signals, and empirically examine whether we can do so with no more than a few examples. Our contributions are as follows:

- We introduce the idea of using LLMs as a proxy reward function.
- We propose a general RL training framework that leverages this proxy reward and is agnostic to the RL algorithm used.
- We show that an LLM can more accurately train objective-aligned RL agents by an average of $35\%$ compared the baseline. We use few-shot prompting for the *Ultimatum Game* and DEALORNODEAL negotiation task as well as zero-shot prompting in *Matrix Games*.
- We conduct a pilot study with 10 human users. Users rate our agent to be significantly more aligned with their objective than an agent trained with a different one, $p < 0.001$.
- We provide further analysis quantifying the amount of user data required for our approach as well as the effect varying prompts has on the LLM's reward signal accuracy.

## 2  RELATED WORK

**Using Language for Reward Shaping**. Recent Reinforcement Learning from Human Feedback (RLHF) works Ouyang et al. (2022); Bai et al. (2022) use LLMs as rewards by fine-tuning them on large amounts of user data. Our work does not fine-tune LLMs but uses in-context learning from only a handful of user data.

Several works Goyal et al. (2019); Carta et al. (2022); Mirchandani et al. (2021) shape rewards by training an RL agent to learn and complete intermediate tasks guided by language. In contrast, our framework does not focus on generating subtasks but leverages the in-context learning abilities of an LLM to determine whether an agent's policy satisfies the higher-level task.

**RL and Foundation Models**. We leverage large language models such as GPT-3 (Brown et al., 2020) to learn a proxy reward function while avoiding the need for many expert demonstrations. Ahn et al. (2022); Huang et al. (2022) use an LLM to provide a plan which guides a robot with reasonable/feasible actions towards a human goal (e.g., with enumerating subtasks). In contrast, our work is different in that we are using an LLM to identify if a behavior satisfies "hard-to-specify" properties of a human's objective and also offers users more control over how they want their policy to be executed. In the vision domain, Parisi et al. (2022) used pre-trained vision models as a feature extractor for the learned policy, but not to design a reward signal. In a similar spirit of leveraging self-supervised pre-training to design a flexible reward function, Chen et al. (2021) use a broad dataset of human videos and a small dataset of robot videos to train a reward function, which improves generalization to new environments and tasks. The interface for the desired task is a video of the task to be completed, instead of text in our framework, and the domain is restricted to robot tasks. More related works can be found in Sec. A.2.

## 3 USING LLMs AS A REWARD SIGNAL

Our goal is to use an LLM as a proxy reward function to train objective-aligned RL agents from user inputs. We formalize the task using a Markov Decision Process $\mathcal{M} = \langle \mathcal{S}, \mathcal{A}, p, \mathcal{R}, \gamma \rangle$, where $\mathcal{S}$ is the state space (e.g., in DEALORNODEAL, the space of representations of utterances in the negotiation so far), $\mathcal{A}$ is the action space (e.g., set of all possible utterances), $p : \mathcal{S} \times \mathcal{A} \times \mathcal{S} \rightarrow [0,1]$ is the transition probability, and $\gamma$ is the discount factor. Traditionally the reward function maps states and actions to a real number $\mathcal{R} : \mathcal{S} \times \mathcal{A} \rightarrow \mathbb{R}$. In our work, we use an LLM as a proxy reward function that takes in a text prompt and outputs a string. We define $A^*$ to be the set of all strings, $\rho \in A^*$ as our text prompt (input to the LLM) and the LLM as a function $LLM : A^* \rightarrow A^*$. As illustrated in Fig. 1, the prompt $\rho$ is a concatenation of four components including a string to describe the task $\rho_1 \in A^*$ and a user-specified string that describes their objectives using examples or a description, $\rho_2 \in A^*$. Additionally, we include a textual description of states and actions from an RL episode, $\rho_3$, using a parser $f : \mathcal{S} \times \mathcal{A} \rightarrow A^*$. $\rho_3$ can describe the final state, final action, a trajectory, or any other representation of the episode. Finally, we include a question $\rho_4 \in A^*$ that asks whether the RL agent's behavior, $\rho_3$, satisfies the user's objective, $\rho_2$. We define an additional parser $g : A^* \rightarrow \{0,1\}$ that maps the textual output of $LLM$ to a binary value. We use this as the reward signal. Our framework replaces the traditional reward function with a proxy reward, $LLM$, and can be used with any RL training algorithm.

Our framework is depicted in Fig. 1. Before training, a user specifies $\rho_2$ which can be $N$ examples describing their objective or a description of their objective using natural language. In Fig. 1 a user provides an example of their objective: versatile negotiating behavior. During training, we construct a prompt $\rho$ by concatenating a description of the task, the user-specified examples/description, an episode's outcome, and a question asking if the outcome satisfies the objective. We (1) feed the prompt to the LLM, (2) take its output, and (3) parse it into an integer using function $g$; we use a handcrafted, task-specific parser. We use the integer as the reward signal. (4) The RL agent then updates its weights and rolls out an episode. (5) We parse the episode outcome into a string using $f$ and continue training; we also instantiate $f$ as handcrafted, task-specific parser. To evaluate our framework, we sample a trajectory (e.g., a negotiation) from the agent and evaluate whether the trajectory is aligned with the user's objective (e.g., whether *Alice* demonstrated versatile negotiating behavior).

## 4 EXPERIMENTS

In this section we investigate three questions to determine the feasibility and efficacy of our approach: (Q1) Can LLMs produce reward signals that are consistent with user objectives from a few examples (few-shot prompting)? (Q2) When objectives are well-known, can LLMs produce objective-consistent reward signals without *any* examples (zero-shot prompting)? (Q3) Can LLMs provide objective-aligned reward signals from examples (few-shot prompting) in more complex, longer-horizon domains? We evaluate our approach on three tasks: the *Ultimatum Game*, 2-player *Matrix Games*, and the DEALORNODEAL negotiation task (Lewis et al., 2017). We address (Q1) using the *Ultimatum Game*. We use *Matrix Games* to address (Q2) because it has well-known solution concepts such as Pareto-optimality. The

DEALORNODEAL negotiation task is a longer-horizon domain where the LLM rewards agents for negotiating in a user-specified style; we address (Q3) in this task.

In practice, we do not have access to ground truth user reward functions — this is the function that we are trying to approximate. However, for most of our experiments, we assume access to the true reward by constructing user reward functions that humans have been shown to have inspired by prior work. *We use the true rewards only to evaluate our framework's performance.* Finally, we include a pilot user study where we evaluate agent performance when we do not have access to the ground truth reward. We use the 'text-davinci-002' GPT-3 model with temperature $0$ as our LLM and our results are reported across $3$ random seeds. Details on how we trained RL agents for each task are in A.4. s

**Evaluation Metrics**. We evaluate our approach using the following metrics across our tasks (task-specific metrics are described within each subsection):

*Labeling Accuracy*. We construct ground-truth reward functions for each domain. We report the mean accuracy of predictions of the reward value *during RL training* with respect to the ground-truth reward functions. This assesses how effectively the LLM can produce reward signals that are consistent with the user's objective.

*RL Agent Accuracy. After RL training*, we evaluate the learned policy with respect to the ground truth reward functions. We report the mean accuracy of RL agents.

**Baselines**.

*SL (Few-shot baseline)*. A supervised learning (SL) model trained to predict reward signals using the same examples given to the LLM in our framework. Examples are represented using structured non-text inputs, making it an easier problem for the SL model. This baseline only applies to tasks where we use few-shot prompting (*Ultimatum Game*, DEALORNODEAL). See A.5 for details on training and model architecture for each task.

*No Objective (Zero-shot baseline)*. In our zero-shot task, *Matrix Games*, we do not use any examples so we do not use SL as a baseline. Instead, we use a *No Objective* baseline where we prompt the LLM without using the user's description of their objective to isolate the effect the description has on the LLM's response.

*RL trained with Ground Truth Reward Functions*. RL agents trained with ground truth reward functions. We use this as an oracle.

## 4.1 ULTIMATUM GAME: TRAINING OBJECTIVE-ALIGNED AGENTS WITH FEW-SHOT PROMPTING

When defining a precise objective is difficult, we can instead give a few examples of desired behavior. For instance, in a resource division game like the *Ultimatum Game*, it may be difficult for a user to specify the exact percentage (such as $32.4\%$) of resources they would be happy with receiving. Instead it could be easier for a user to give examples of splits that they would be happy with. We explore whether LLMs can produce reward signals that are consistent with user objectives from a few examples in the *Ultimatum Game*.

**Task Description**. The *Ultimatum Game* consists of two players, a Proposer and a Responder. A sum of money is endowed to the Proposer and they must propose a way to split the endowment with the Responder. The Responder can accept or reject the proposed split. If the Responder accepts, players receive money as per the split; if the Responder rejects, then both players get nothing. We train an RL agent to play the Responder. The agent learns to reject proposals according to a user's preferences. The game consists of a single timestep and our RL agents are trained using DQN for $1e4$ steps.

**Ground Truth User Objectives**. A rational Responder would accept any proposal, even if it is unfair because getting something is better than getting nothing (in fact, this is a Nash Equilibrium of the game). However, prior work in behavioral economics shows that humans are willing to "punish" the Proposer by rejecting unfair proposals (Vavra et al., 2018). For instance, a student may reject an unfair proposal only if she receives less than 30% of the endowment whereas a wealthier person may reject if they receive less than 60%. We experiment with the following preferences:

- **Low vs High Percentages.** Users will reject proposals if they receive less than $\{30\%, 60\%\}$ of the endowment.
- **Low vs High Payoffs.** Users will reject unfair proposals if they receive less than $\{\$10, \$100\}$. They accept unfair proposals otherwise.

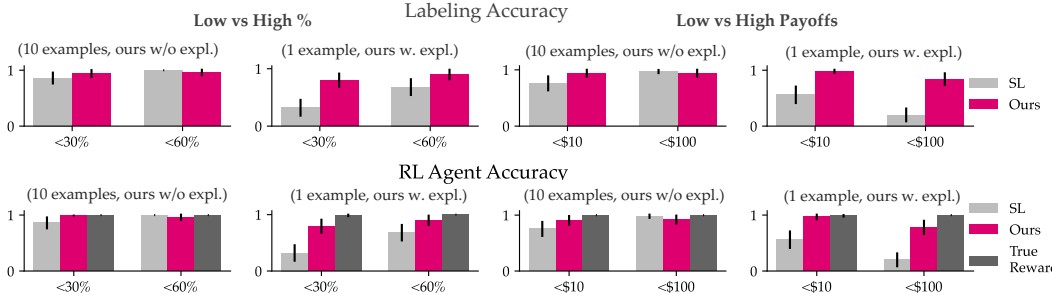

Figure 2: **Ultimatum Game, Few-shot**. (Top) Accuracy of reward signals provided by LLM and SL during RL training when prompted with/trained on 10 vs 1 example. (Bottom) Corresponding accuracy of RL agents after training. LLM is able to maintain a high accuracy when prompted with a single example followed by an explanation. We do not provide figures of *Inequity Aversion* because both LLM and SL trivially achieve perfect labeling and RL agent accuracy.

- **Inequity Aversion (Fehr & Schmidt (2010)).** Users will reject proposals if they do not receive exactly $50\%$ of the endowment.

**Prompt Design**. We describe a user's objective using 10 examples of the *Ultimatum Game*. An example consists of the proposed split, the Responder's action, and a "yes/no" label of whether the Responder's action was desirable or undesirable. These examples do not have explanations and resemble a traditional dataset used for supervised learning. We also experiment with using a single example followed by a short explanation. Importantly, we do not explicitly mention the user's ground truth objective in the prompt. See Fig. 10 in the Appendix for an example of both types of prompts.

*Design Procedure.* We randomly generated 10 proposed splits used for our prompt and sampled one proposal from the set for our single-example case. For *Low vs High Percentages* and *Low vs High Payoffs*, we used the same set of proposals across variants (i.e., same proposals for (30% and 60%) and ($10, $100)). We also randomly generated 50 proposals used to evaluate the LLM. Due to limited resources when querying GPT-3, we query the model's responses to the 50 evaluation splits in a batched manner and save them. We then use those responses as the reward signal.

### 4.1.1 RESULTS

**Labeling Accuracy**. We evaluated our approach on our test set of 50 proposals over 3 seeds; results are shown in Fig. 2[1]. When prompted with 10 examples without explanations, the LLM and SL perform similarly well (see Fig. 2, top row). This result is not surprising, given that the decision boundary for the binary decision tasks is relatively simple to learn with 10 training examples.

Instead, if we prompt the LLM with a single example followed by an explanation, it maintains a high accuracy whereas SL trained on the same, single example drops in accuracy. We did not use the explanation as part of input when training SL because it only takes non-textual inputs. This result highlights the advantage of using an LLM over a supervised learning model: they require far fewer examples because they can learn from *explanations* (Lampinen et al., 2022). We find that explanations are critical, as removing explanations when prompting the LLM with a single example results in a drop in LLM labeling accuracy (avg. drop of $31.67\%$) and a drop in RL agent accuracy (avg. drop of $28.8\%$).

**RL Agent Accuracy**. The accuracy of the trained RL agents mirror the labeling accuracy.

***Summary***. *LLMs are efficient in-context learners. They are able to provide reward signals that are consistent with a user's objectives from examples — even a single example with an explanation will suffice.*

### 4.2 MATRIX GAMES: TRAINING OBJECTIVE-ALIGNED AGENTS WITH ZERO-SHOT PROMPTING

When objectives are well-known concepts such as Pareto-optimality, can we prompt the LLM without giving any examples? We hypothesize that well-known objectives are likely to be in-distribution for LLMs, and thus LLMs may be able to produce objective-aligned reward signals from zero-shot prompting. Since

---

[1]We do not display plots for *Inequity Aversion* because both LLM and SL received perfect labeling and RL agent accuracy.

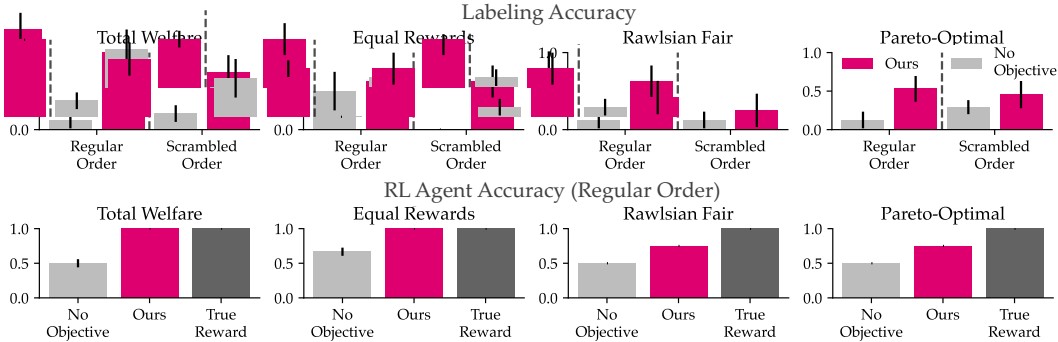

Figure 3: **Matrix Games, Zero-shot**. (Top) Accuracy of reward signals provided by LLM and a *No Objective* baseline during RL training. We report results for both regular and scrambled versions of matrix games. (Bottom) Accuracy of RL agents after training.

we do not use examples, we do not use a SL baseline. Instead we use a baseline *No Objective* where we do not mention any objectives and ask the LLM for a reward signal (see example in Fig. 11 in the Appendix). This baseline evaluates whether the LLM can successfully apply its knowledge of each objective.

**Task Description**. We consider two-player normal-form matrix games: Battle of the Sexes, Stag Hunt, Chicken, and Prisoner's Dilemma. Each matrix game has four joint outcomes (i.e., a tuple of joint actions and rewards) and we address pure strategies in this task. The game consists of a single timestep and our RL agents are trained using DQN for 500 steps.

**Ground Truth User Objectives**. Although (mixed) Nash Equilibria are traditional solution concepts for normal form matrix games, users may prefer a solution for other properties. For instance, in Prisoner's Dilemma, users may prefer both agents to cooperate because they will maximize total welfare even though it is not a Pure Nash Equilibrium. We experiment with four well-known solution concepts (or objectives):

- **Total Welfare.** Outcomes that achieve the greatest sum of player rewards.
- **Equality.** Outcomes that result in equal rewards between players.
- **Rawlsian Fairness.** Outcomes that maximize the minimum reward any player receives.
- **Pareto-optimality.** Outcomes where the one of the corresponding rewards cannot be improved without lowering the other.

**Prompt Design**. Prompts for each solution concept are shown in Fig. 11 in the Appendix. Due to limited resources with querying GPT-3, we queried GPT-3 in a batched manner and saved the corresponding labels to train our RL agents. Our prompt enumerates the outcomes of a matrix game and then asks the LLM for the outcome(s) that satisfy a solution concept. We do not mention the name of the matrix game in the prompt. As in Kojima et al. (2022), we elicit intermediate reasoning steps by asking the LLM to "think step-by-step" and provide a definition of the solution concept. To prevent any bias the LLM may have towards the order in which the outcomes of a matrix game are presented, we also randomly scramble associations between joint actions and rewards (example shown in Fig. 12 in the Appendix).

*Design Procedure.* We tuned the wording of our prompt (e.g., how to describe the matrix game, whether or not to use chain-of-thought prompting) on the Battle of the Sexes matrix game to find a prompt that gave us accurate results. During evaluation, we kept the structure of our prompt the same for all of the matrix games.

### 4.2.1    RESULTS

**Labeling Accuracy**. Given that each game can have many outcomes that satisfy a solution concept, we report the LLM's accuracy if its response does not include any incorrect outcomes. If the LLM identifies any incorrect outcome, we report a score of 0. The LLM produces more objective-aligned reward signals with zero-shot prompting by applying its knowledge of well-known objectives, improving the labeling accuracy over having no objective by 48% on average with a regular ordering of matrix game outcomes and 36% with a scrambled order. Scrambling the order of matrix game outcomes in the prompt lowers accuracy for most solution concepts. We suspect that this is because the matrix games are well-known and likely to have been in the LLM's training set, where each joint action is usually associated with particular

payoffs. Scrambling the associations between joint actions and payoffs could make the matrix game more out-of-distribution for the LLM, and thus lower accuracy.

**RL Agent Accuracy**. Compared to labeling accuracy, it is easier for the resulting RL agents to be accurate because they only need to learn *one* correct outcome, not all of them. Thus, LLMs that only identify one out of two correct outcomes can still train objective-aligned RL agents. Results are shown on the bottom row of Fig. 3. RL agents trained using rewards from the LLM receive perfect accuracy for *Total Welfare* and *Equality* and 75% accuracy for the other two objectives. The baseline receives lower accuracy for all objectives.

**Does the LLM Correctly Identify Each Objective?** As part of our prompt, we ask the LLM to provide a definition of the objective before reasoning about whether an outcome satisfies the objective (see Fig. 11 in the Appendix). Table 2 in the Appendix shows how the LLM defines each objective zero-shot. The LLM is able to successfully recall the definitions for each objective except for *Rawlsian Fairness* – it gets it partially correct. However, the LLM varies in its ability to reason whether an outcome of a game satisfies the objective, which explains why the LLM does not receive perfect labeling accuracy for all objectives.

*Summary*. *An LLM is able to identify well-known objectives and provide objective-aligned reward signals in a zero-shot setting.*

### 4.3 DEALORNODEAL: TRAINING OBJECTIVE-ALIGNED AGENTS IN MULTI-TIMESTEP TASKS

We have shown that an LLM can provide objective-aligned reward signals in single-timestep tasks. In longer horizon tasks we must give trajectories instead of states as examples in our prompts. Longer prompts can be challenging because it is less likely for an LLM to have seen them during training. LLMs also have a recency bias which makes it harder for them to remember context introduced earlier on (Zhao et al., 2021). Can an LLM provide objective-aligned signals in longer horizon tasks? We investigate this question in the DEALORNODEAL negotiation task (Lewis et al., 2017).

**Task Description**. DEALORNODEAL is a long-horizon task with a maximum length of 100 timesteps. An agent *Alice* must come to an agreement with her partner *Bob* on the allocation of a set of objects (*books*, *hats*, and *balls*). Agents are shown a context, which includes the counts of each item and their private utilities for each item. In the original task, agents get rewarded based on the agreed upon split and their utilities. If *Alice* and *Bob* reach a disagreement, both agents get nothing. We train *Alice* using on-policy RL by negotiating against a fixed partner model, which we refer to as *Bob*. See Sec. A.4 for more details on the domain and training.

**Ground Truth User Objectives**. We train *Alice* to negotiate in different *styles*. For this experiment, we assume we have access to precise definitions in order to evaluate our models. Importantly, we do not give definitions of each style to the LLM, only examples. We experiment with the following negotiation styles inspired by previous literature Sycara et al. (1997); Caputo et al. (2019):

- **Versatile.** *Alice* does not suggest the same proposal more than once.
- **Push-Over.** *Alice* gets less points than *Bob*.
- **Competitive.** *Alice* gets more points than *Bob*.
- **Stubborn.** *Alice* repeatedly suggests the same proposal.

**Prompt Design**. We describe user objectives using three examples. Each example contains a negotiation between *Alice* and *Bob*, a question asking whether *Alice* negotiated in a particular style, and a yes or no answer followed by a short explanation. For an example, see Fig. 13 in the Appendix.

*Design Procedure*. To create example negotiations, we randomly sampled three negotiation contexts for each objective and trained an RL agent (using the original task reward, without the LLM) to negotiate against *Bob* in these contexts. We then sampled negotiations from the trained model. We also made sure all three sampled negotiations did not have the same ground truth label. We use a separate set of contexts when training RL agents with an LLM in the loop.

#### 4.3.1 RESULTS

**Labeling Accuracy**. The top row of Fig. 4 shows that the LLM labels more accurately than SL except for *Versatile*. For *Versatile*, both models perform similarly because SL learns to overwhelmingly predict a negative label (avg of 96% negative predictions) and the RL agent showed more negative examples of *Versatile* behavior (avg. of 70% ground truth negative labels). However, the large portion of negative

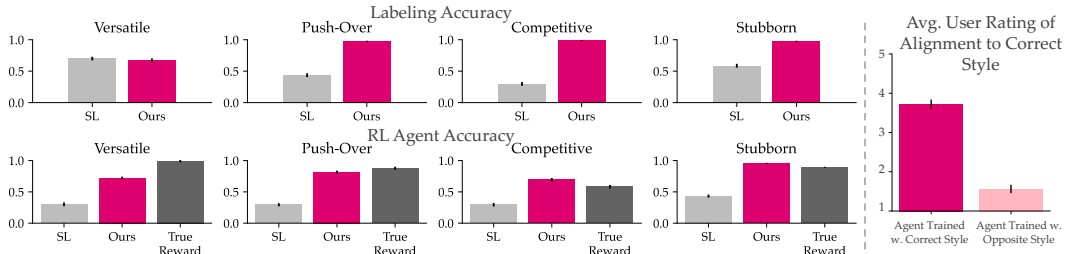

Figure 4: **DEALORNODEAL, Few-shot**. (Top) Accuracy of reward signals provided by LLM and SL during RL training. (Bottom) Accuracy of RL agents after training. (Right) Pilot study results. Agents trained with the user's preferred style were rated as significantly more aligned than an agent trained with the opposite style $p < 0.001$.

|  | Advantage | Diversity | Agreement Rate |
|---|---|---|---|
| Versatile | $0.17 \pm 0.91$ | $0.99 \pm 0.01$ | $0.98 \pm 1.89$ |
| Push-Over | $-2.95 \pm 0.64$ | $0.82 \pm 0.26$ | $1.0 \pm 0.0$ |
| Competitive | $2.92 \pm 0.64$ | $0.74 \pm 0.25$ | $0.88 \pm 6.5$ |
| Stubborn | $1.36 \pm 2.24$ | $0.52 \pm 0.1$ | $0.82 \pm 12.35$ |

Table 1: Qualitative results describing negotiations produced by agents trained with LLM.

examples prevents the agent from learning correct behavior as shown in the *Versatile* plot on the bottom of Fig. 4); here, we get a larger performance gap between the LLM and SL.

**RL Agent Accuracy**. Results are shown on bottom row of of Fig. 4. LLM improves RL agent accuracy over SL by $46\%$ on average. Our method approaches the performance of using the true reward; we under perform by an average of $4\%$. We remind readers that it *is* possible to outperform an agent trained with the true reward — especially when the LLM's labeling accuracy is near-perfect as is the case for *Competitive* and *Stubborn* — due to reward hacking or stochasticity during training. For instance, agents trained with the true reward for *Competitive* end in more disagreements, leading to $0$ reward for both agents and a lower accuracy.

**Is There a Qualitative Difference in Styles?** Example negotiations of agents trained with the LLM for each style are shown in Fig. 9 in the Appendix. To measure qualitative differences among styles, we looked at average advantage (Alice's original task reward - Bob's original task reward), diversity (percentage of Alice's utterances that are unique in a negotiation), and agreement rate (percentage of negotiations that end in agreement). The results in Table 1 demonstrate qualitative differences in styles.

### 4.3.2 PILOT USER STUDY

We conduct a within-subjects pilot user study to determine whether our trained agents can meaningfully align themselves with different user objectives *when we do not have access to ground truth objectives* and *users evaluate agent performance*.

**Method** We asked $N = 10$ users to select a style in which they wanted their agent to negotiate in. We gave them an option to choose from our existing styles (*Versatile*, *Push-Over*, *Competitive*, *Stubborn*), or come up with their own. Importantly, users did not know how we defined these styles. We then showed users a list of 10 example negotiations generated via selfplay using an RL agent trained with a greedy objective (no particular style). We asked users to select 3 examples (1 positive, 1 negative, and 1 positive or negative) where Alice displayed positive or negative behavior of the user's chosen style. For each chosen example, we asked users whether *Alice* demonstrated their chosen style and asked them to provide a "Yes/No" answer as well as a short explanation. These examples corresponded to unknown, user-specific ground truth reward functions that we did not have access to. We then trained a negotiation agent by incorporating the user-provided examples in the prompt as described in Sec. 3. We also trained an agent to negotiate in the opposite style by flipping the "Yes/No" labels in the user-provided explanations. We hypothesize that users should perceive a significant difference between these two agents. To evaluate our trained agents, we had both agents negotiate on a set of 10 test negotiations. For each test negotiation, we asked users to rate how well each agent aligned with their chosen style on a scale from 1 (least aligned) to 5 (most aligned).

**Results** Agents trained with the correct style were significantly more aligned (avg. $3.72 \pm 1.2$) than agents trained with the opposite style (avg. $1.56 \pm 1.05$), $p < 0.001$, see Fig. 4 (right). Users varied in which

styles they preferred: 4 users chose styles such as *Polite*, *Push-Over*, *Considerate* and *Compromising*, 2 users chose *Versatile*, and 4 users chose styles such as *Stubborn*, *Competitive* and *Ambitious*. These results demonstrate that our framework can produce agents aligned with differently specified objectives by changing the examples in the prompt. Furthermore, these results suggest that our framework can be used when rewards are difficult to define and results are agents with humans.

*Summary*. *Our framework can train objective-aligned agents when ground truth rewards are not present in complex, longer-horizon tasks. Agents are able to align the style in which they complete a task as evaluated by automated metrics as well as human users.*

## 5 ANALYSIS OF DATA EFFICIENCY & PROMPT DESIGN

We have shown that we can use an LLM as a proxy reward function to successfully train objective-aligned agents across different tasks. This is a promising result because it represents an important step in enabling human-compatible and value-aligned AI systems. In this section, 1) we further quantify how data efficient our method is and 2) also analyze how robust LLM is to variations of prompt design.

**1) How Data-efficient is Our Method?** We quantify how much *more* user data a supervised learning baseline would need in order to achieve the same labeling accuracy as the LLM. Results in the *Ultimatum Game* demonstrate that 10 labeled examples is enough for an SL model to reach comparable performance, whereas a a single labeled example is not sufficient. In this section, we quantify the amount of data needed for DEALORNODEAL because it is an example of a task where the decision boundary is not as easy to learn as the *Ultimatum Game*. We train SL by adding additional class-balanced, labeled examples to the original three examples it was trained on. We plot the average labeling accuracy SL achieves when trained on increasingly larger amounts of examples as well as the accuracy LLM achieves with three examples, shown in Fig. 6 in the Appendix. *Results show that SL requires on the order of hundreds of more labeled examples in order to be comparably accurate as LLM.*

**2) How Much Does LLM's Labeling Accuracy Change When We Vary the Prompt?** As with many approaches that use LLMs, a limitation of our approach is that it requires prompt design. We attempt to quantify the effort required for designing prompts as well as determine the feasibility of using non-engineered prompts from humans. We analyze the effect of prompt variation on labeling accuracy in DEALORNODEAL for the *Stubborn* objective. We vary different parts of the user-specified prompt at a time: the keyword (i.e., replacing "Stubborn" with its synonyms), the example negotiations, and the explanations associated with each example. Fig. 7 provides a summary of our results. See Sec. A.7 for the full results. Results illustrate that an LLM can be quite robust to different prompts — they all outperform SL. Furthermore, the quality of the explanation seems to be the most important in determining LLM accuracy.

## 6 LIMITATIONS & FUTURE WORK

**User Studies**. This work takes a first step in determining whether we can use LLMs as proxy rewards. Given promising results, we plan on evaluating our approach with a larger user study.

**Multimodal Foundation Models**. Beyond language models, multimodal foundation models such as Flamingo (Alayrac et al., 2022) can enable us to provide more complex environment states to the foundation model through images or other modalities while preserving an intuitive language interface for specifying the user objective.

**Non-Binary Rewards**. Another limitation of our framework is that the LLM only specifies binary rewards. We plan on exploring how we can incorporate the likelihoods that LLMs produce for each word as a non-binary reward signal.

ACKNOWLEDGMENTS

This work was supported by NSF Award 1941722, 2125511, 2006388, AFOSR, DARPA YFA Award, ONR, and JP Morgan Faculty Award. We would also like to thank Karl Tuyls, Ian Gemp, Albert Gu, Siddharth Karamcheti, Kanishk Gandhi, and other reviewers for this paper.

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

# A APPENDIX

## A.1 SUMMARY

### OF RESULTS: IS IT IS POSSIBLE TO USE LLM AS A PROXY REWARD IN RL TRAINING?

| | Avg. Labeling Accuracy | | | Avg. RL Agent Accuracy | | | |
|---|---|---|---|---|---|---|---|
| | Zero-shot Baseline (No Obj.) | Few-shot Baseline (SL) | Ours | Zero-shot Baseline (No Obj.) | Few-shot Baseline (SL) | Ours | True Reward |
| *Ultimatum Game* | -- | $0.67 \pm 0.34$ | $\mathbf{0.91 \pm 0.27}$ | -- | $0.67 \pm 0.34$ | $\mathbf{0.9 \pm 0.26}$ | $\mathbf{1.0 \pm 0.02}$ |
| *Matrix Games* | $0.19 \pm 0.29$ | -- | $\mathbf{0.61 \pm 0.41}$ | $0.54 \pm 0.29$ | -- | $\mathbf{0.88 \pm 0.}$ | $\mathbf{1.0 \pm 0.}$ |
| DEALORNODEAL | -- | $0.5 \pm 0.47$ | $\mathbf{0.9 \pm 0.22}$ | -- | $0.33 \pm 0.42$ | $\mathbf{0.8 \pm 0.33}$ | $\mathbf{0.84 \pm 0.27}$ |

Figure 5: Average Labeling and RL Agent Accuracy across the different objectives for each task across 3 seeds.

We provide a summary of results in Fig. 5. The figure depicts the average Labeling and RL Agent Accuracy computed across the different user objectives for each task, across 3 seeds. Overall, our approach is able to produce more objective-aligned reward signals than our baselines. Our approach is also able to produce objective-aligned policies that are close in accuracy to policies trained with the true reward.

## A.2 MORE RELATED WORKS

**Reward Design**. Our framework addresses reward design—how to engineer rewards so that they align with our objectives (Amodei et al., 2016). This is challenging because tasks often have conflicting objectives that a human must trade off (Pan et al., 2022). Misspecifying reward functions can lead to *reward hacking*, or the gaming of specified rewards. Reward hacking has appeared in various domains such as autonomous driving (Knox et al., 2021) and game-playing (Ibarz et al., 2018). We hope to address these challenges by leveraging LLMs and making it easier for humans to specify their objectives.

**Imitation Learning & Preference-based Learning**. Another method of specifying user objectives is to learn them from expert demonstrations (Ross et al., 2011) or preferences Sadigh et al. (2017). These techniques either assume access to large datasets (Christiano et al., 2017) or place restrictive assumptions (such as linearity) about the reward function (Sadigh et al., 2017). Recent work attempts to learn reward functions from language instructions using pragmatic reasoning (Lin et al., 2022). In contrast, our work relies on an LLM's in-context learning abilities to provide a reward.

## A.3 LLM DEFINITION OF OBJECTIVES IN THE MATRIX GAME

| **LLM Defns. of Objectives** |
|---|
| Total welfare is the sum of the rewards of both players. (✓) |
| Equality of rewards is only possible if both players receive the same reward. (✓) |
| Rawlsian fairness is defined as the maxmin value of the game, which is the minimum reward that the player could get assuming that the other player is maximizing their reward. (X) |
| An outcome is Pareto-optimal if there is no other outcome that would make one player better off without making the other player worse off. (✓) |

Table 2: Completion of each sentence given by LLM in pink. LLM provides correct definitions for objectives except for *Rawlsian Fairness*, which is partially correct.

## A.4 DETAILS ON RL ENVIRONMENTS AND TRAINING

**Ultimatum Game**. The environment is a single horizon with discrete actions (i.e., accept or reject) and continuous observations (i.e., a proposed split). We train DQN agents using the Stable Baselines3 implementation for 1e4 timesteps with a learning rate of 1e-4 across 3 seeds (Raffin et al., 2021). We instantiate our policy as a MLP with the default parameters used in Stable Baselines3.

*Parser g.* The parser $g$ that transforms the LLM's output into an integer reward signal is defined using a handcrafted parser. When prompting the LLM, we structure the labels for each example to be in "Yes/No" form which enables the LLM to also reply using the same format. We are then able search for the "Yes" or "No" strings and parse them into a 1 or 0 respectively. In the rare occasion that the LLM does not respond in this form, we skip the episode during RL training and omit the example from our evaluation.

*Further Analysis on Performance of* No Objective *Baseline.* The average LLM accuracy of a random baseline across the four matrix games are *Welfare*: 0.125, *Equality*: 0.125, *Rawlsian Fairness*: 0.078, *Pareto-optimality*: 0.172. The *No Objective* baseline's performance is close to random as can be verified by comparing the random baseline results with Figure 3 (top row).* Behaviorally, we observe that *No Objective* acts like a random baseline: the LLM often hallucinates matrix game rewards and also displays incoherent reasoning when selecting answers.

**Matrix Game.**. The environment is a single horizon with discrete actions (i.e., one of the four joint actions) and no observations. We train DQN agents using the Stable Baselines3 implementation for 500 timesteps with a learning rate of 1e-4 across 3 seeds (Raffin et al., 2021). We instantiate our policy as a MLP with the default parameters used in Stable Baselines3.

*Parser g.* We parse the LLM's response by hand, since LLM output can be variable in zero-shot settings.

**DEALORNODEAL**. We use a version of the DEALORNODEAL environment used in Kwon et al. (2021). In the environment, the goal is for an agent $A$ to come to an agreement with a partner $B$ on the allocation of a set of objects (*books*, *hats*, and *balls*). During each negotiation, agents receive a *context*, $c_A = [i; u_A], c_B = [i; u_B]$, detailing the count of each item $i$ as well as their private utilities, $u_A, u_B$. Item counts and utilities are represented as vectors $i \in \{1,...,4\}^3$ and $u_A, u_B \in \{0,...,10\}^3$ and are sampled uniformly.

After receiving contexts $c_A, c_B$, an agent is randomly selected to begin the negotiation. Agents negotiate for $T$ time steps by exchanging coarse dialogue acts $x_t$ at each time step $1 \leq t \leq T$ (He et al., 2018). Rather than negotiate directly in natural language, where the generation problem is hard and can result in degenerate dialogues (He et al., 2018), we use these dialogue acts instead to focus on learning diverse and interpretable strategies.

A dialogue act $x_t$ is one of five actions: `propose`, `insist`, `agree`, `disagree`, or `end`. The `propose` and `insist` acts take allocations of items as arguments $o = [o_A; o_B]$ where $o_A, o_B \in \{1,...,4\}^3$ (e.g., `propose: books=1, hats=2, balls=1`). When an agent selects `end`, the conversation terminates and each agent is asked to make their final selection.

If agents agree on the final allocation of items, i.e., $o_A + o_B = i$, agents are awarded points based on their private utilities, $r_A = u_A \cdot o_A, r_B = u_B \cdot o_B$. If agents do not agree, they receive 0 points. Each agent's context is constrained so that the agent can receive a maximum of 10 points.

Agents are first trained using supervised learning on a dataset of human-human negotiations provided by (Lewis et al., 2017) to predict the next token. We use a learning rate of 1.0 and batch size of 16. We then fine-tune these agents using RL where they optimize the expected reward of each dialogue act using REINFORCE (Williams, 1992). Agents are trained on 250 contexts for 1 epoch with a learning rate of 0.1. We instantiate our policy with four GRUs (Chung et al., 2014). We closely follow the implementation outlined in (Kwon et al., 2021; Lewis et al., 2017), please refer to those papers for more training details.

*Parser g.* The parser $g$ that transforms the LLM's output into an integer reward signal is defined using a handcrafted parser. When prompting the LLM, we structure the labels for each example to be in "Yes/No" form which enables the LLM to also reply using the same format. We are then able search for the "Yes" or "No" strings and parse them into a 1 or 0 respectively. In the rare occasion that the LLM does not respond in this form, we skip the episode during RL training and omit the example from our evaluation.

## A.5 SL MODEL ARCHITECTURE AND TRAINING

**Ultimatum Game**. SL is trained to predict binary labels for a batch of proposed splits. We implemented SL as a multi-layer perceptron (MLP) network that consists of a single hidden layer with depth 32. We also use ReLU activations after our input and hidden layers. We trained the model on the same 10 examples we gave to LLM for 5 epochs with the Adam optimizer. We evaluate the model on the 50 heldout test examples and save the model with the best test accuracy. We show the training and test accuracy for each user objective below:

Table 3: Training accuracy for SL on the *Ultimatum Game*.

| | 30% | 60% | $10 | $100 | Ineq. Aversion |
|---|---|---|---|---|---|
| Train Acc. (10 examples) | 1.0 | 1.0 | 0.9 | 1.0 | 1.0 |
| Train Acc. (1 example) | 1.0 | 1.0 | 1.0 | 1.0 | 1.0 |

**DEALORNODEAL.** SL is trained to predict binary labels given a negotiation as input. We closely follow the implementation of a SL model found in (Kwon et al., 2021; Lewis et al., 2017). A negotiation consists of a context, coarse dialogue acts exchanged between *Alice* and *Bob*, and the outcome of the negotiation (the final split of items and whether agents agreed or disagreed). Please refer to Sec. A.4 for more details on the environment. We implement SL using a MLP context encoder, MLP outcome encoder, and a GRU ((Chung et al., 2014)) to process the coarse dialogue acts. The MLP encoders consist of an embedding layer followed by a linear layer with a Tanh activation function; we use a hidden size of $64$ for the context encoder's linear layer. We similarly embed each coarse dialogue act before feeding it into the GRU. We use a hidden size of $128$ for the GRU. We train SL on the same $3$ examples we use in our prompt for LLM. We train for a maximum of $50$ epochs using the Adam optimizer. SL received a training accuracy of $100\%$ for all of our objectives.

## A.6 HOW DATA-EFFICIENT IS OUR METHOD?

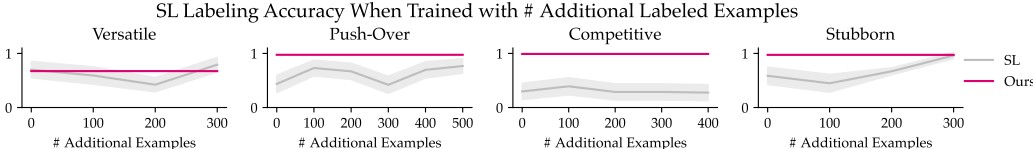

Figure 6: SL requires on the order of hundreds of more labeled examples in order to be comparably accurate to the LLM.

## A.7 HOW MUCH DOES LLM'S LABELING ACCURACY CHANGE WHEN WE VARY THE PROMPT?

Effect of Prompt Variation on Labeling Accuracy for *Stubborn* (N=3, 3 seeds)

| SL | Ours | Vary Keyword | Vary Example Negotiations | Vary Explanations |
|---|---|---|---|---|
| $0.58 \pm 0.48$ | $0.97 \pm 0.16$ | $0.91 \pm 0.28$ | $0.93 \pm 0.28$ | $0.79 \pm 0.33$ |

Figure 7: On average, varying the prompt does not have a large impact on the accuracy of the LLM.

We analyze the effect of varying prompts on the LLM's labeling accuracy for the *Stubborn* negotiating style in DEALORNODEAL. We vary prompts in three ways: we vary the keyword (i.e., replacing "Stubborn" with its synonyms), the example negotiations, and the explanations associated with each example.

When varying the keyword, we use the synonyms: "Headstrong", "Obstinate", and a less commonly used word, "Froward". To vary example negotiations, we randomly sample three new negotiations to have counterbalanced labels, all positive labels, or all negative labels. We vary the explanations by coming up with two plausible sets of explanations a user might have given for each example. We also experiment with the scenario where we give no explanations. Results are shown in Fig. 8. Overall, varying prompts do not have a large impact on labeling accuracy — they all outperform the baseline. However, the quality of explanation seems to have the largest impact on labeling accuracy.

## A.8 WHAT IS THE IMPORTANCE OF INCLUDING THE TASK DESCRIPTION, $\rho_1$ IN THE PROMPT?

We experiment with removing $\rho_1$, the task description in the Ultimatum Game with a single example followed by an explanation. Performance increases slightly in LLM labeling accuracy (avg. of $8\%$) and RL agent accuracy (avg. of $9\%$). We run the same experiment in the Ultimatum game in the case of $10$ examples with no explanation. Performance drops slightly in LLM labeling accuracy (avg. $4.4\%$) and RL

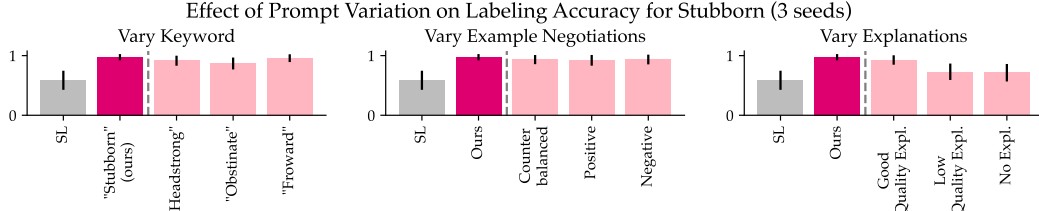

Figure 8: Varying prompts do not have a large impact on labeling accuracy.

agent accuracy (avg. $5.3\%$). We conclude that $\rho_1$ is not conclusively influential in improving performance in few-shot settings.

## A.9 EXPERIMENTING WITH SMALLER LLM SIZES

We experiment with GPT-2, a $1.5B$ parameter model. We We find that GPT-2 underperforms GPT-3 in both labeling (avg. $15\%$) and RL agent accuracy (avg. $49\%$). GPT-2 outperforms the SL baseline in labeling accuracy (avg. $24\%$), and slightly underperforms the SL baseline for RL agent accuracy (avg. $2.7\%$). Results are averaged across styles and seeds. GPT-2 ($1.5B$) is several orders smaller than GPT-3 ($175B$), and we expect models larger than GPT-2 to close the gap with GPT-3's performance.

## A.10 EXAMPLE NEGOTIATIONS

**Context**

Alice : book=(count:1 value:0) hat=(count:1 value:7) ball=(count:3 value:1)
Bob   : book=(count:1 value:3) hat=(count:1 value:7) ball=(count:3 value:0)

**Versatile**
```
-----------------------------------------------------------------
Alice : propose: item0=1 item1=1 item2=0
Bob   : propose: item0=0 item1=1 item2=0
Alice : propose: item0=1 item1=0 item2=2
Bob   : agree
-----------------------------------------------------------------
Agreement!
Alice : 2 points
Bob   : 7 points
```

**Push-over**
```
-----------------------------------------------------------------
Alice : propose: item0=1 item1=0 item2=3
Bob   : propose: item0=1 item1=1 item2=0
Alice : agree
-----------------------------------------------------------------
Agreement!
Alice : 3 points
Bob   : 10 points
```

**Competitive**
```
-----------------------------------------------------------------
Alice : propose: item0=1 item1=0 item2=3
Bob   : propose: item0=1 item1=1 item2=0
Alice : insist: item0=0 item1=1 item2=3
Bob   : agree
-----------------------------------------------------------------
Agreement!
Alice : 10 points
Bob   : 3 points
```

**Stubborn**
```
-----------------------------------------------------------------
Alice : propose: item0=0 item1=1 item2=1
Bob   : propose: item0=0 item1=1 item2=0
Alice : propose: item0=0 item1=1 item2=1
Bob   : propose: item0=0 item1=1 item2=0
Alice : propose: item0=0 item1=1 item2=1
Bob   : propose: item0=0 item1=1 item2=0
Alice : propose: item0=0 item1=1 item2=1
Bob   : propose: item0=0 item1=1 item2=0
Alice : propose: item0=0 item1=1 item2=1
Bob   : propose: item0=0 item1=1 item2=0
Alice : propose: item0=0 item1=1 item2=0
Bob   : disagree
-----------------------------------------------------------------
Disagreement?!
Alice : 0 (potential 0)
Bob   : 0 (potential 0)
```

Figure 9: Example negotiations after *Alice* is trained with reward signals from LLM in DEALORNODEAL. We illustrate qualitative differences in how *Alice* negotiates for the same context. *Bob* is an agent that is trained with supervised learning.

## A.11 EXAMPLE OF PROMPTS USED IN OUR EXPERIMENTS

### A.11.1 ULTIMATUM GAME

*Further Explanation of Our Prompt Selection Process.* When constructing our explanations, we encourage the LLM to produce intermediate reasoning steps by using the "Let's think step by step" template used in Kojima et al. (2022) which has been shown to improve performance.

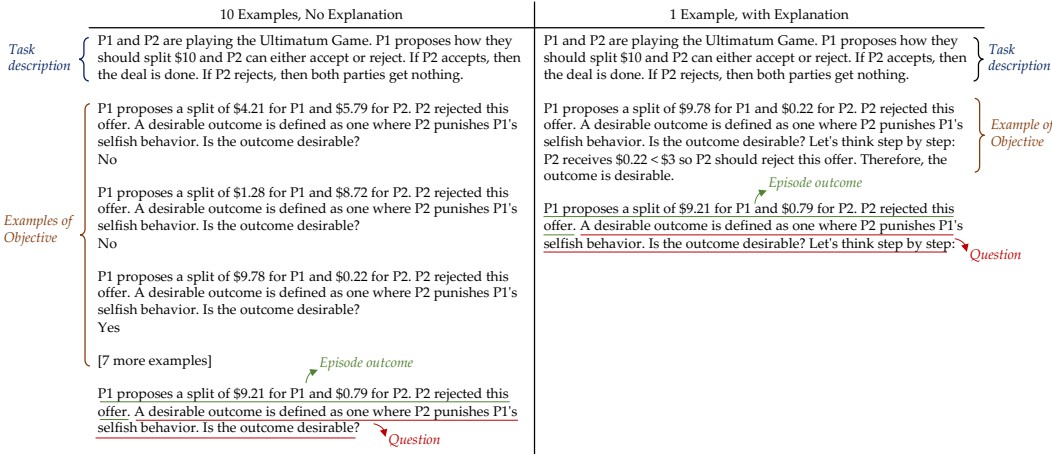

Figure 10: An example of few-shot prompts used for the *Ultimatum Game*. We highlight the four parts of each prompt.

### A.11.2 MATRIX GAMES

*Further Explanation of Our Prompt Selection Process.* We found that structuring the outcomes of the game as a multiple choice question improved performance. We also encouraged the LLM to produce intermediate reasoning steps by using the "Let's think step by step" template used in Kojima et al. (2022) which has been shown to improve performance.

| | Total Welfare | Equality | Rawlsian Fairness | Pareto-optimality | No Objective |
|---|---|---|---|---|---|
| Task description | We have a two-player game where P1 and P2 can choose one of these options. Options: A. if action1(P1) and action1(P2) => P1 gets reward of 2, P2 gets reward of 2. B. if action1(P1) and action2(P2) => P1 gets reward of 1, P2 gets reward of 3. C. if action2(P1) and action1(P2) => P1 gets reward of 3, P2 gets reward of 1. D. if action2(P1) and action2(P2) => P1 gets reward of 0, P2 gets reward of 0. | We have a two-player game where P1 and P2 can choose one of these options. Options: A. if action1(P1) and action1(P2) => P1 gets reward of 2, P2 gets reward of 2. B. if action1(P1) and action2(P2) => P1 gets reward of 1, P2 gets reward of 3. C. if action2(P1) and action1(P2) => P1 gets reward of 3, P2 gets reward of 1. D. if action2(P1) and action2(P2) => P1 gets reward of 0, P2 gets reward of 0. | We have a two-player game where P1 and P2 can choose one of these options. Options: A. if action1(P1) and action1(P2) => P1 gets reward of 2, P2 gets reward of 2. B. if action1(P1) and action2(P2) => P1 gets reward of 1, P2 gets reward of 3. C. if action2(P1) and action1(P2) => P1 gets reward of 3, P2 gets reward of 1. D. if action2(P1) and action2(P2) => P1 gets reward of 0, P2 gets reward of 0. | We have a two-player game where P1 and P2 can choose one of these options. Options: A. if action1(P1) and action1(P2) => P1 gets reward of 2, P2 gets reward of 2. B. if action1(P1) and action2(P2) => P1 gets reward of 1, P2 gets reward of 3. C. if action2(P1) and action1(P2) => P1 gets reward of 3, P2 gets reward of 1. D. if action2(P1) and action2(P2) => P1 gets reward of 0, P2 gets reward of 0. | We have a two-player game where P1 and P2 can choose one of these options. Options: A. if action1(P1) and action1(P2) => P1 gets reward of 2, P2 gets reward of 2. B. if action1(P1) and action2(P2) => P1 gets reward of 1, P2 gets reward of 3. C. if action2(P1) and action1(P2) => P1 gets reward of 3, P2 gets reward of 1. D. if action2(P1) and action2(P2) => P1 gets reward of 0, P2 gets reward of 0. |
| Question | Which option(s) result in the greatest total welfare? Let's think step by step: *Description of Objective* Total welfare is | Which option(s) result in equality of rewards? Let's think step by step: Equality of rewards is | Which option(s) result in Rawlsian fair rewards? Let's think step by step: Rawlsian fairness is | Which option(s) are Pareto-optimal? Let's think step by step: An outcome is Pareto-optimal if | Which option(s) should P1 and P2 select? |

Figure 11: Examples of a zero-shot prompts used for each objective, including the no-objective baseline, in the *Matrix Games*. Due to limited resources when querying GPT-3, we queried GPT-3 in a batched manner and saved the corresponding labels to train our RL agents. Consequently, we do not have an *Episode outcome* in our prompts.

### A.11.3 DEALORNODEAL

*Further Explanation of Our Prompt Selection Process.* We chose 3 counterbalanced examples from a training set of sample negotiations. This training set was generated via selfplay using an RL agent trained

| Chicken Game (Regular Order) | Chicken Game (Scrambled Order) |
|---|---|
| We have a two-player game where P1 and P2 can choose one of these options.
Options:
A. if action1(P1) and action1(P2) => P1 gets reward of 2, P2 gets reward of 2.
B. if action1(P1) and action2(P2) => P1 gets reward of 1, P2 gets reward of 3.
C. if action2(P1) and action1(P2) => P1 gets reward of 3, P2 gets reward of 1.
D. if action2(P1) and action2(P2) => P1 gets reward of 0, P2 gets reward of 0.

Which option(s) are Pareto-optimal? Let's think step by step:
An outcome is Pareto-optimal if | We have a two-player game where P1 and P2 can choose one of these options.
Options:
A. if action1(P1) and action1(P2) => P1 gets reward of 3, P2 gets reward of 1.
B. if action1(P1) and action2(P2) => P1 gets reward of 2, P2 gets reward of 2.
C. if action2(P1) and action1(P2) => P1 gets reward of 1, P2 gets reward of 3.
D. if action2(P1) and action2(P2) => P1 gets reward of 0, P2 gets reward of 0.

Which option(s) are Pareto-optimal? Let's think step by step:
An outcome is Pareto-optimal if |

Figure 12: Examples of regular and scrambled outcomes for the Chicken Game (*Matrix Game* experiments). Scrambling creates new associations between joint actions and joint rewards. We scramble the outcomes in order to remove any bias LLM may have towards the order in which outcomes are normally presented.

with a greedy reward (no particular style). We chose to complement our examples with simple and succinct explanations.

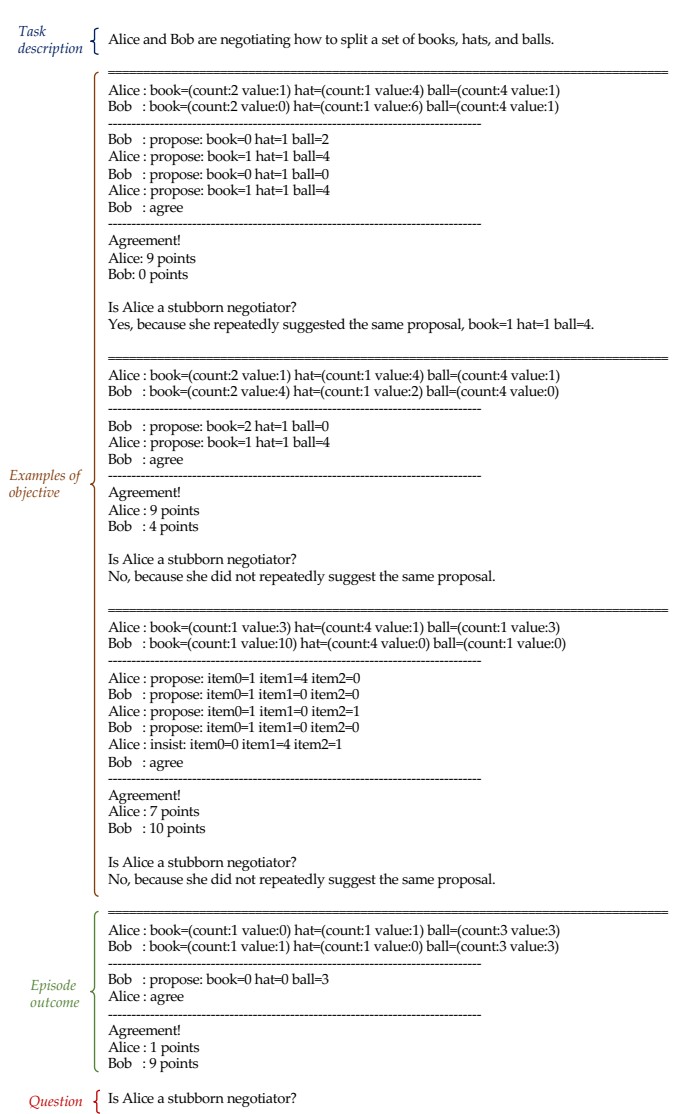

Figure 13: Example of a prompt used for DEALORNODEAL.

