# OpenReview forum: "Reward Design with Language Models"
_ICLR.cc/2023/Conference — ICLR 2023 poster_

### Official Review · Reviewer_49Yr · 2022-10-20

**Confidence:** 5
**Correctness:** 3
**Technical Novelty And Significance:** 3
**Empirical Novelty And Significance:** 3
**Recommendation:** 8

**Clarity, Quality, Novelty And Reproducibility:**

Clarity: Very good
Quality: The paper is above-average to high quality.
Novelty: I believe this idea is novel.
Reproducibility: Some RL training details are sparse, but LLM prompts are given. Somewhat reproducible.

**Strength And Weaknesses:**

## Strengths

****************************************Writing and Clarity:****************************************

- The writing is very intuitive and straightforward; the paper is easy to read.

******************************Framing/Method:******************************

- This is a great execution of a simple idea (simple being a good thing here). The authors tested reward design with LLM on a variety of environments and train an RL-agent in-tandem to prove that not only are the labels accurate with the LLM, but these labels allow RL agents to learn well on tasks compared to the true underlying reward functions.

************************Experiments:************************

- The authors have comprehensive experiments demonstrating one of the claims of the paper: that LLM’s can learn from far fewer examples due to all the data they’re trained on.

## Weaknesses

**Clarity:**

- Some details regarding the games seem to be missing, for example what is the fixed Bob agent in DealOrNoDeal?

**Experiments:**

- Missing LLM size comparison: While using GPT-3 helps demonstrates the points the authors aim to make about the ability of LLMs to act as reward functions, GPT-3 `text-davinci-002` and even open source alternatives of the same size are prohibitive to most researchers either due to cost (GPT) or compute limitations (not enough VRAM to run inference). This also presents a problem because the experiments don’t really show that LLMs in general can help design reward functions, instead it shows that a 176B param GPT-3 can help design reward functions. Can the authors experiment with smaller alternatives? For example, at least GPT-3 `curie` (a 1-line code change for the authors), and/or the OPT family of open-sourced models (e.g. 350m, 1.3b, 6.7b - compute-matched with `curie`, 30b).

******************************************Missing related work:******************************************

- Using Natural Language to aid with reward design is a related subfield with some works, here is one for example: [https://www.ijcai.org/proceedings/2019/0331.pdf](https://www.ijcai.org/proceedings/2019/0331.pdf). This is a little bit related to the framing, as the authors should probably add a related works section explaining language as a proxy for rewards.

****************************Minor details:****************************

- With temperature 0 isn’t GPT-3 just producing the max-likelihood token? In that case I don’t believe top-p changes anything about the model’s outputs and therefore is irrelevant (Section 4 2nd paragraph).
- Unfinished Appendix: A.5 and A.6 are not used and don’t have any text.
- Typos:
    - “They require far few examples” > “far fewer”
    - “on the agreed upon the agreed upon split”

**Summary Of The Paper:**

This paper proposes using LLMs to understand underlying reward functions given a few language examples, then demonstrate that RL agents trained with this LLM-provided sparse reward function nearly match those trained with true underlying reward functions on some simple tasks.

**Summary Of The Review:**

I like this paper, and it demonstrates a well-written execution of a simple idea. The main issue that I have with this paper is the lack of experiments on smaller or more accessible LLMs. With that said, I still believe this paper should be accepted.

---

> ### Author Response · Authors · 2022-11-12
> **Experiment on LLM Size Comparison**
>
> We thank 49Yr for the insightful review. We were excited that 49Yr notes that the paper “is a great execution of a simple idea” with “comprehensive experiments”. We address individual questions below:
>
> > “GPT-3 text-davinci-002 and even open source alternatives of the same size are prohibitive to most researchers either due to cost (GPT) or compute limitations (not enough VRAM to run inference). Can the authors experiment with smaller alternatives? ”
>
> We thank 49Yr for the suggestion to include LLM size comparison results. During the rebuttal period, we experimented with GPT2 (1.5B), a model that is free and accessible via Hugging Face, in the DealOrNoDeal domain. We find that GPT2 underperforms GPT3 in both labeling (avg. 15%) and RL agent accuracy (avg. 49%). GPT2 outperforms the SL baseline in labeling accuracy (avg. 24%), and slightly underperforms the SL baseline for RL agent accuracy (avg. 2.7%). Results are averaged across styles and seeds. GPT2 (1.5B) is several orders smaller than GPT3 (175B), and we expect models larger than GPT2 to close the gap with GPT3’s performance. We have added this result to Section A.11 in the Appendix.

---

> > ### Comment · Reviewer_49Yr · 2022-11-13
> > **Updating Score**
> >
> > After looking at the other reviews, author responses, and new draft, I believe this paper is improved from the original one during submission. The authors have responded appropriately to all reasonable questions/experiment requests.
> >
> > I still believe that this paper is appropriate for this conference and well-motivated. I am raising my score .

---

> ### Author Response · Authors · 2022-11-12
> **Missing Related Work and Other Minor Comments**
>
> >”Using Natural Language to aid with reward design is a related subfield with some works, here is one for example: https://www.ijcai.org/proceedings/2019/0331.pdf. This is a little bit related to the framing, as the authors should probably add a related works section explaining language as a proxy for rewards.”
>
> We thank 49Yr for the suggestion. We have added a “Using Language For Reward Shaping” paragraph in Sec. 2 of the paper and have also included the citation. In this new section, we discuss prior works that use language to help RL agents navigate sparse reward tasks by providing intermediary feedback (e.g., providing information about subtasks via language). We have included other works such as ELLA: Exploration Through Learned Language Abstraction (2021) and EAGER: Asking and Answering Questions for Automatic Reward Shaping in Language-guided RL (2022).
>
> >”With temperature 0 isn’t GPT-3 just producing the max-likelihood token? In that case I don’t believe top-p changes anything about the model’s outputs and therefore is irrelevant (Section 4 2nd paragraph).”
>
>  Thank you for pointing this out. You are absolutely is correct and we have omitted the description of the top-p parameter in our paper.
>
> >”Unfinished Appendix: A.5 and A.6 are not used and don’t have any text.”
>
>  A.5 and A.6 are blank due to formatting issues (our Appendix has a lot of large images), which we apologize for. We have fixed this in the paper.
>
> >List of typos
>
> We thank 49Yr for catching these typos and have fixed them.

---

### Official Review · Reviewer_7igZ · 2022-10-21

**Confidence:** 3
**Correctness:** 1
**Technical Novelty And Significance:** 3
**Empirical Novelty And Significance:** 4
**Recommendation:** 8

**Clarity, Quality, Novelty And Reproducibility:**

**Clarity:**
- Generally a very clear and well written paper, as stated in the strength sections. I like how each game is designed to test one of the hypothesis and the summary at the end of each experiment with an explanation of the findings.
- The example in Figure 1 could be improved. For instance you could replace item0... for books/hats/balls, and specify the total number, or how many go for Alice and How many for Bob.
- The sentence "we assume access to the true reward by constructing user reward functions that humans have been shown to have inspired by prior work", is a bit confusing. Are the rewards based on prior work? While this is further clarified in Sec 4.1 GT User objectives, an example of such reward here would be helpful.

- Clarification Question: In 4.2.1 the authors seem to be describing recall on correct outcomes (identifying correct outcome) but I imagine that they are actually measuring accuracy of outcomes? If so they should modify the text.
- Comment: What is the role of the description \rho1, when prompts in \rho2 are given.
- Minor:
    - Moving Figure 2 to page 5 would make the results a bit easier to read.
    - Because SL actually does not have the explanation, I would modify the title in Figure 2, and add explanation in the Ours bar.

**Novelty**:
- To my knowledge, this is the first approach using LLMs to specify rewards, it is thus a novel formulation and a promising one, as LLMs become a building block of applications in ML.

**Reproducibility**:
The domains and rewards are thoroughly explained in the appendix. More description of the process for selecting the right prompts, and especially the natural language explanations for the reward would be appreciated. It would also be valuable if authors released the dataset used for prompting the language model. As long as `text-davinci-002` remains available the results should be reproducible but given the importance of good explanations as per figure 6, it it important that authors share the dataset of explanations to ensure reproducibility.


**Strength And Weaknesses:**


**Strengths:**

- Clear and well written paper. The goal of the paper is clear and well motivated, backed by prior work evidence, the contributions of the paper are clearly stated, the method is also clearly written, even though a bit more clarity could be added into the particular example in figure 1. The experimental design, results and takeaways are clear as well.
- The approach for reward specification is flexible, allowing both example-based specification, through a description of the desired task or a combination of the 2.
- To my knowledge, this is the first approach using LLMs to specify rewards, it is thus a novel formulation and a promising one, as LLMs become a building block of applications in ML.
- Good experimental design. While the tasks are relatively simple, they represent scenarios for which we care about specifying rewards, and prove the 3 main points that authors are claiming with the proposed metthod.
- Very interesting to see that the LLM contains concepts such as pareto-optimality. It would be great to see if the LLM can combine multiple concepts at the same time, when possible, or negate concepts. I don't think it is a requirement for this paper but I would suggest adding it as follow up work.
- As shown in 5.2, the method is relatively robust to different prompts, as long as an explanation for the reward is given, implying that the method should not necessarily require thorough prompt design.

**Weaknesses:**

- Given that one of the motivations in this work is to provide rewards in cases where it hard to engineer them, I think that this paper would strongly benefit from having a case where the reward cannot be formally described, and measure with a human study whether people prefer policies that come from rewards generated from the LLM vs policies that come from people playing successful games according to such reward.

- Experiments:
    - I am missing more results to understand the role of explanations vs few-shot. In experiment 1, what is the performance if we use a LLM with only one example and no explanation? Does the high performance come solely from explanation or it is also because the model has better capacity for few-short learning. On that line, what is the importance of \rho1? Does the LLM perform the task worse when given few shot examples and no description of the task?

    - I am missing more analysis and context on some of the results. In 4.2.1 what is the performance of a random baseline? Is there bias towards a given user objective when it is not given, or in the failure cases, or are the best case solutions random?



- While this is also the case of works like Huang et al. 2022, a limitation of this approach is that it requires states and actions in the environment to be represented as tokens that the LLM has been trained on. This limitation is transparently addressed in the last section though, and since many tasks can actually be specified via language, I still think this work is a valuable contribution. A minor suggestion would be that besides commenting on multi-modal states, authors also mentioned non-token actions. This may be important for continuous control for instance, where we may not want aggressive motion in a robot arm.

- See my comments on clarity.

**Summary Of The Paper:**

This paper proposes to use a pre-trained large language model to specify rewards for reinforcement learning. The model takes a description of the task and reward, and optionally a small set of demonstrations and outputs a binary reward for a new queried game state. Based on this reward, an RL agent is trained to complete the desired task and is compared to agents trained with the GT reward as well as a reward generated by training a supervised model on the task demonstrations.
Through experiments in different domains, authors show that: 1) The proposed model can represent rewards, and train successful RL agents from even a single example, 2) LLMs can use background knowledge and predict correct rewards from commonly known objectives, even in a zero-shot setting and 3) They can model rewards in long horizon tasks, provided demonstrations and explanations.


**Summary Of The Review:**

As stated in my strengths section, the paper addresses a very important problem with a solution that is simple and relatively general-purpose (safe for some of the cases mentioned in the limitations). The contributions are very clear and the design of the games and experiments show that using language models to specify rewards offers advantages in multiple ways. Moreover, as LLMs become a more pervasive building block in decision-making models, the approach has the potential to be extended and generalized as more research is done on this end (for instance through multi-modal foundation models to address more general states). As such, I think this is a valuable paper for this conference. The reason why I cannot give a strong accept is because I am missing experiments at the core of this paper goal: providing rewards where there is no way to specify them, evaluating them using human ratings.  Moreover, I would really like to clarify some of the details in the experimental results, as stated in the weakness section, experiments. This would help me convince that the proposed approach offers an effective way to specify rewards.

[EDIT]: The authors have addressed all my concerns, clarifying experimental design and metrics, adding thorough ablation studies and a human experiment that proves the effectiveness of the approach under real scenarios with hard to specify rewards. I recommend this paper for acceptance.

---

> ### Author Response · Authors · 2022-11-12
> **We Include a Pilot User Study where Users Find Our Agents to Be Well-Aligned.**
>
> We thank Reviewer 7igZ for their detailed and insightful comments. We were excited that the reviewer thought  “the paper addresses a very important problem”, a “novel formulation and a promising one”, with “good experimental design”. We address individual questions below:
>
> > “I think that this paper would strongly benefit from having a case where the reward cannot be formally described, and measure with a human study whether people prefer policies that come from rewards generated from the LLM”
>
> As 7igZ suggests, during the rebuttal phase, we conducted a within-subjects pilot user study (N=10) in the DealOrNoDeal task. **We focus this study to determine whether our trained agents can meaningfully align themselves with different user objectives when the objectives are hard to define and users rate agent performance. Users find our agents to be significantly well-aligned compared to agents trained with a different objective, p < 0.001.** We have added results to Section 4.3.2 of the paper.
>
> We asked 10 users to pick a style in which they wanted their agent to negotiate in. Then we ask the users to provide positive and negative examples for their style by selecting example negotiations from a library of different negotiation behaviors. From this library, the user selects 3 negotiation behaviors, where Alice does or does not demonstrate their style (with short explanations). *Importantly, we had no ground truth access to what these styles should be defined as, making it difficult to engineer any type of reward.* We then trained a negotiation agent using the user-provided examples and explanations.
>
> To illustrate how effectively our framework can produce aligned agents, we also trained an agent in the opposing style by flipping the “Yes/No” labels of the user-provided explanations. We should expect users to perceive a large difference between the two agents.
>
> We asked each user to rate (from 1 - least aligned to 5 - most aligned) how aligned both negotiation agents are to their preferred style on 10 test negotiations.  Agents trained with the correct style were significantly more aligned (avg. 3.72 \pm 1.2) than agents trained with the opposite style (avg. 1.56 \pm 1.05), p < 0.001. This result suggests that our framework is truly versatile and is able to produce agents aligned with differently specified objectives by changing the explanations in the prompt. More details on the methods of this study can be found in Sec. 4.3.2.

---

> ### Author Response · Authors · 2022-11-12
> **Results on Questions Regarding Experiments**
>
> 7igZ asks great clarifying questions regarding our experiments. We apologize if these were not clear in our first draft.  We have now provided additional results and clarifications below and have added them to the paper.
>
> >”I am missing more results to understand the role of explanations vs few-shot. In experiment 1, what is the performance if we use a LLM with only one example and no explanation? Does the high performance come solely from explanation or it is also because the model has better capacity for few-short learning.”
>
> As 7igZ suggests, during the rebuttal period, we experiment with using an LLM with only one example with no explanation in the Ultimatum Game. **We find that removing explanations results in a drop in LLM labeling accuracy (avg. drop of 31.67%) and a drop in RL agent accuracy (avg. drop of 28.8%).** We posit that without explanations, a single example is not expressive enough to capture the user’s full objective. We have edited the paper to explain this (Sec. 4.1.1).
>
> >”On that line, what is the importance of \rho1? Does the LLM perform the task worse when given few shot examples and no description of the task?”
>
> As 7igZ suggests, during the rebuttal period we find that **$\rho_1$ is not conclusively influential in improving performance in few-shot settings and have edited the paper to acknowledge this (Sec. A.7 in the Appendix).** We experiment with removing $\rho_1$, the task description in the Ultimatum Game with a single example followed by an explanation. Performance actually increases slightly in LLM labeling accuracy (avg. of 8%) and RL agent accuracy (avg. of 9%). We run the same experiment in the Ultimatum game in the case of 10 examples with no explanation. Performance drops slightly in LLM labeling accuracy (avg. 4.4%) and RL agent accuracy (avg. 5.3%).
>
> >”I am missing more analysis and context on some of the results. In 4.2.1 what is the performance of a random baseline? Is there bias towards a given user objective when it is not given, or in the failure cases, or are the best case solutions random?”
>
> The average LLM accuracy of a random baseline across the four matrix games are Welfare: 0.125, Equality: 0.125, Rawlsian Fairness: 0.078, Pareto-optimality: 0.172. **The “No Objective” baseline does not have a bias towards a particular user objective— its performance is close to random as can be verified by comparing the random baseline results with Figure 3 (top row).** Behaviorally, we observe that “No Objective” acts like a random baseline: the LLM often hallucinates matrix game rewards and also displays incoherent reasoning when selecting answers. We have edited the paper to clarify this point (Sec. A.4 in the Appendix).

---

> ### Author Response · Authors · 2022-11-12
> **Other Comments on Clarity and Reproducibility**
>
> > “The example in Figure 1 could be improved. For instance you could replace item0... for books/hats/balls, and specify the total number, or how many go for Alice and How many for Bob.”
>
> We thank 7igZ for the suggestions and edited Figure 1 accordingly.
>
> >”Clarification Question: In 4.2.1 the authors seem to be describing recall on correct outcomes (identifying correct outcome) but I imagine that they are actually measuring accuracy of outcomes? If so they should modify the text.”
>
> We are measuring accuracy only for answers that do not contain any incorrect options. We have clarified the text in Sec. 4.2.1.
>
> > “Minor: Moving Figure 2 to page 5 would make the results a bit easier to read.” and “Because SL actually does not have the explanation, I would modify the title in Figure 2, and add explanation in the Ours bar.”
>
> We have made the suggested changes to Fig. 2
>
> > “More description of the process for selecting the right prompts, and especially the natural language explanations for the reward would be appreciated.”
>
> We agree this is an important component when using LLMs, and have added more description of the prompt selection process in Section A.10 of the Appendix.
>
> > “It would also be valuable if authors released the dataset used for prompting the language model.”
>
> We have added a ‘Reproducibility’ section in the paper (Sec. 7) and have uploaded code for generating the exact prompts used in our experiments as well as code for training agents as part of the Supplementary Materials.

---

### Official Review · Reviewer_UKq1 · 2022-10-21

**Confidence:** 3
**Correctness:** 3
**Technical Novelty And Significance:** 2
**Empirical Novelty And Significance:** 2
**Recommendation:** 5

**Clarity, Quality, Novelty And Reproducibility:**

My major concern about this paper is its novelty. While it may be quite interesting for the community, I feel that its findings are not sufficient for the level of ICLR.

**Strength And Weaknesses:**

Strengths:
- Using textual prompts to drive RL agents is an emerging topic with promising outcomes

- Promising results for controling RL agents with language

Weakness:

- Using pre-trained LLMs in zero-shot settings is clearly not new, many studies already exist on that topic. Not on the specific tasks authors suggest, but the transferability of their embedded knowledge with well specified prompts is well known. Prompt tuning is an emerging field following that observation.

- Shaping rewards with LLMs is not new either, as many works recently proposed to leverage them to split difficult tasks into textual subgoals, or to drive agents by asking questions for instances, in a curriculum RL fashion. These works should at least be mentioned in the related work.

- Writing is not always very clear. Readers have to go in many various places again and again during reading to get things together. Very difficult to understand without appendix. Many settings not sufficiently formally specified to well analyse results (e.g., the supervised learning of SL, the definition of function g, etc.).

- Environments look quite simple (short trajectories, few actions, deterministic transitions, full observability). It would be helpful to see more classical complex tasks of RL, in environments such as TextWorld, Alfred or BabyAI for instances.

**Summary Of The Paper:**

The paper "Reward Design with Language Models" proposes to study to effectiveness of pre-trained large language models for designing rewards in RL tasks (tasks implying language). It shows that, somehow unsurprisingly given recent literature, leveraging these models to drive agents with textual prompts is possible in few-shot or zero-shot settings.

**Summary Of The Review:**

  - Could authors position their work w.r.t literature on prompt tuning, zero-shot learning with pre-trained LLMs (in particular, the last reference I give below looks very connected to the presented work), and also w.r.t. reward shaping with language?
   - The global learning setting is not fully clear to me. How is defined or learned the function g that maps the textual outcome of the LLM to an int reward ? It is a language model too ? It is learned from binary true rewards ? If it is manually defined, how do authors manage variability of llm outputs ? (for expected answers such as yes or no, it is possible that the llm outputs something different right ? or do authors restrict its possible outcomes ?)
   - In fig 8 right, the prompt finishes with "let think step by step :". So the LLM is supposed to propose a step by step forecasting of reasoning ?





Multitask Prompted Training Enables Zero-Shot Task Generalization. ICLR 2022

Finetuned language models are zero-shot learners. arXiv preprint arXiv:2109.01652.

Ella: Exploration through learned language abstraction. In Advances in Neural Information Processing Systems (NeurIPS), 2021

EAGER: Asking and Answering Questions for Automatic Reward Shaping in Language-guided RL. CoRR abs/2206.09674 (2022)

Do as i can, not as i say: Grounding language in robotic affordances. arXiv preprint arXiv:2204.01691. (2022)

---

> ### Author Response · Authors · 2022-11-12
> **Added Relevant Literature; Discussion of Why Our Approach is Novel (Post 1/2)**
>
> > Q1. UKq1 states that “Using pre-trained LLMs in zero-shot settings is clearly not new”, “Shaping rewards with LLMs is not new either”, and asks that the “authors position their work w.r.t literature on prompt tuning, zero-shot learning with pre-trained LLMs… and also w.r.t. reward shaping with language?”, citing the following related works to add:
> * Multitask Prompted Training Enables Zero-Shot Task Generalization. ICLR 2022
> * Finetuned language models are zero-shot learners. arXiv preprint arXiv:2109.01652.
> * Ella: Exploration through learned language abstraction. In Advances in Neural Information Processing Systems (NeurIPS), 2021
> * EAGER: Asking and Answering Questions for Automatic Reward Shaping in Language-guided RL. CoRR abs/2206.09674 (2022)
> * Do as i can, not as i say: Grounding language in robotic affordances. arXiv preprint arXiv:2204.01691. (2022)
>
> We thank the reviewer for their comments –- we believe there may be a misunderstanding with regards to novelty, and we provide a thorough comparison to the related works mentioned by the reviewer below.
>
> **Novelty:** We address Reviewer UKq1’s main concerns regarding the novelty of our approach. We were surprised by the reviewer’s comment with respect to novelty, as we are actually very excited about the novelty of our work, as to the best of our knowledge, our approach of using LLMs for reward design in the context of reinforcement learning was not studied before and can be a promising direction for future reward design and reinforcement learning challenges. In addition, **we’d like to point out that all other reviewers agree that our approach is novel**:
> * “As far as I know, this approach is new” (1rSQ),
> * “To my knowledge, this is the first approach using LLMs to specify rewards, it is thus a novel formulation and a promising one, as LLMs become a building block of applications in ML” (7igZ),
> * “I believe this idea is novel” (49Yr).

---

> ### Author Response · Authors · 2022-11-12
> **Added Relevant Literature; Discussion of Why Our Approach is Novel (Post 2/2)**
>
> **Comparison to mentioned works:** We thank UKq1 for the list of relevant works — we have added a ‘Using Language for Reward Shaping’ paragraph in Sec. 2 of the paper to include the suggested citations. We provide a summary of how our work relates to the suggested citations:
>
> * Multitask Prompted Training Enables Zero-Shot Task Generalization (2021)
> * Finetuned Language Models Are Zero-Shot Learners (2021)
>
> These two works are not very related to our setting. Specifically both works are focused on fine-tuning LLMs on large downstream datasets to improve zero-shot performance on natural language tasks. In contrast, we are not concerned with improving LLM performance, but rather *using*  LLMs to improve RL agent performance. Our work focuses on the problem of reward specification and we train RL agents that are *aligned* with human values using LLMs as a reward. Our work also does not require large datasets (it is actually data efficient, see Section 5).
>
> * ELLA: Exploration Through Learned Language Abstraction (2021)
> * EAGER: Asking and Answering Questions for Automatic Reward Shaping in Language-guided RL (2022)
>
> Both works tackle the sparse reward problem by training an RL agent to learn and complete relevant subtasks to accomplish a higher-level task guided by language. While these works are also concerned with training RL agents, they take a very different approach and are mainly applicable when subtasks/subgoals can be found through exploration. This is not always easy, e.g., it is not clear what subtasks to explore in the context of negotiation games we’re interested in. In contrast, our framework does not focus on how to generate subtasks and is not limited to the “goal-conditioned” settings, but instead it leverages the novel idea of querying an LLM to determine whether an agent’s policy satisfies the higher-level task – tapping into all the context available to LLMs for assessing this. However, we would like to note that our framework can be used in conjunction with this line of works.
>
> * Do as i can, not as i say: Grounding language in robotic affordances. arXiv preprint arXiv:2204.01691. (2022)
>
> Finally, with respect to the SayCan work above, we would like to first note that we already had a discussion of this work in Sec. 2 of the submission draft. This work is quite different as they do not address the reward design problem or the reinforcement learning problem. In a similar vein as ELLA and EAGER, this work also attempts to solve the problem of instruction following by breaking it down into subtasks, and it leverages an LLM for the “breaking down into subtasks part”. However, the way we’re using the LLM is fundamentally different: we are using an LLM to identify if a behavior satisfies “hard to specify” properties, e.g. “if a round of negotiations was fair” instead of asking an LLM to output the subtasks of a longer horizon objective. This allows us to use LLMs as a way of evaluating reward functions and address the challenging problem of reward design.
>
> We further use the outcome of LLM in the context of reinforcement learning, allowing us to have much more control on the policy that is being learned (e.g., *how* the robot cleans a room). This is in contrast with the SayCan work, which outputs a high-level task plan only checking for feasibility of the high level task and relying on a library of low-level hand-coded motion primitives
>
> We originally had a section describing the differences between our approach and this work. However, we have now expanded that to emphasize and clarify these points:
>
> > Ahn et al. (2022); Huang et al. (2022) use an LLM to provide a plan which guides a robot with reasonable/feasible actions towards a human goal(e.g., with enumerating subtasks). In contrast, our work is fundamentally different in that we are using an LLM to identify if a behavior satisfies ``hard-to-specify'' properties of a human's objective and also offers users more control over how they want their policy to be executed.
>
> We also included a discussion of two other works at the intersection of Foundation Models and RL/reward design in Section 2 of the submission draft:
> >In the vision domain, Parisi et al. (2022) used pre-trained vision models as a feature extractor for the learned policy, but not to design a reward signal. In a similar spirit of leveraging self-supervised pre-training to design a flexible reward function, Chen et al. (2021) use a broad dataset of human videos and a small dataset of robot videos to train a reward function, which improves generalization to new environments and tasks. The interface for the desired task is a video of the task to be completed, instead of text in our framework, and the domain is restricted to robot tasks.

---

> ### Author Response · Authors · 2022-11-12
> **Clarifying Other Details of the Paper**
>
> > Q2. “How is defined or learned the function g that maps the textual outcome of the LLM to an int reward?” and “how do authors manage variability of llm outputs ?”
>
> In our experiments, we define g using a handcrafted parser. For our few-shot tasks (Ultimatum Game, DealOrNoDeal), we manage variability in LLM responses by structuring the example responses in our prompt to be in the form “Yes, [explanation]” or “No, [explanation]”. This enables the LLM to give responses in the same manner and allows us to parse the response into an integer by looking at the first word. In the rare occasion when the LLM does not give a response in this manner, we skip the episode during RL training and omit the example in our evaluation. (This happened approximately less than 5 times across 3000 episodes). It is harder to write a parser for our zero-shot task (Matrix Games), and thus we parsed the LLM’s answers by hand. We thank UKq1 for this comment and clarified how we define g in Sec. A.4 of the Appendix.
>
> > Q3. “In fig 8 right, the prompt finishes with "let think step by step :". So the LLM is supposed to propose a step by step forecasting of reasoning ?”
>
> This prompt improves the performance of language models in reasoning-based tasks, and improves the LLM’s outputs in our approach. Our prompts incorporate ideas from work on eliciting reasoning capabilities from language models, such as the use of “Let's think step by step''[1] where we provide intermediate steps towards the answer in the few-shot examples.
>
>
> [1] Kojima et al., (2022) Large Language Models are Zero-Shot Reasoners

---

### Official Review · Reviewer_1rSQ · 2022-10-26

**Confidence:** 3
**Correctness:** 3
**Technical Novelty And Significance:** 2
**Empirical Novelty And Significance:** 3
**Recommendation:** 5

**Clarity, Quality, Novelty And Reproducibility:**

**Originality**

The authors provide an interesting approach to solve text based RL tasks. As far as I know, this approach is new.

**Clarity**

I feel the authors should clarify and explore the pros and cons of the approach, for both discussion and concrete experiments. It is not enough to show the limitations at the final conclusion section, since the the limitation of your approach I feel is some very fundamental problems to study and may help others to understand.

**Quality**

Overall the paper is well written and the experiments shows the effectiveness of their approach on the limited domains. The quality can be further improved if the authors could answer my previous questions.

**Strength And Weaknesses:**

**Strength**
- The idea is simple and intuitive, which is quite easy to understand.
- The authors conducted detailed ablation study to show the effective of using LLM as Reward


**Weakness**

- I believe the approach can only be used for text based games, can not used for other more general tasks such as control tasks.
- The authors only evaluate three tasks, and two of the are just 1 step tasks, and the left one is multi-step horizon based, but from the description it seems that the horizon is not that long.
- Also, from the illustration figure, I feel the current approach may only deal with short horizons, since you will need to consider the whole trajectory as the prompt input to the LLM.
- There is no user study. Since your tasks involves languages, I feel it should be necessary to conduct some user study with human evaluation, thus we can get some sense of what exactly happens here.
- Also, for RL researchers, people would usually have the impression that  sparse reward would not actually work in practice, and require some more techniques to improve. Can authors explain why sparse reward provided by your approach actually work? What's the difference if we use a sparse intermediate reward function for training? Will that actually work?

**Summary Of The Paper:**

The authors in this paper study how to utilize large language models (LLM) as a proxy reward function for text based RL such as Ultimatum game, matrix game or the DealOrNoDeal negotiation tasks. The main idea is to consider to take a combined prompt $\rho$ as the input of LLM, which consists of the task description, user provided example, converted outcome of an episode, and  a question asking if the outcome episode satisfies the user objective, and then use. the LLM's response as signals for reward.

**Summary Of The Review:**

Overall I think the paper introduce an interesting approach for utilizing reward function for text based RL tasks, though the current approach is not quite general and can only applied  to short horizon based methods, and there is no user study with human evaluation to understand what is really going on here.

---

> ### Author Response · Authors · 2022-11-12
> **Our Method Generalizes to Control Tasks, Longer Horizon Tasks, and Non-Sparse Rewards**
>
> We thank the reviewer for the valuable feedback. 1rSQ notes that “the idea is simple and intuitive” and the “approach is new”. Reviewer 1rSQ raised valid concerns as to whether our approach will generalize to 1) control tasks 2) longer-horizon tasks and 3) non-sparse rewards. We clarify how our method generalizes to each of these factors below:
>
> > Q1. 1rSQ asks whether our approach “can only be used for text based games, can not used for other more general tasks such as control tasks”
>
> Non-text-based games can absolutely be used with our approach. For environments where state-action spaces cannot be easily converted into language (e.g., training a robot to stack blocks), one can featurize the state-action space with higher-level descriptions (e.g., “the robot picked up the block”). Melting Pot [1] is an example of a suite of environments whose state-action space has been featurized and can potentially benefit from our approach.
>
> If we consider replacing the LLM with any other multi-modal Foundation Model (such as Vision-Language Models (VLM)) we do not have to represent control tasks using language, but instead images/videos (or some other modality). This is exciting and a promising avenue with the new advancements and research in multi-modal Foundation Models.  Once we’ve replaced the LLM with a VLM, we can use the exact same training procedure in our paper. We believe that these are exciting areas for future work and will enhance the generality of our training procedure.
>
> Finally, there are many environments where the state-action space can be easily converted into language because they are simple enough (e.g., simple gridworld games), or because there exists a dataset of accompanying language descriptions (e.g., MineCraft [2]). Our method is immediately applicable to any of these domains.
>
> > Q2. 1rSQ asks whether our “current approach may only deal with short horizons, since you will need to consider the whole trajectory as the prompt input to the LLM.”
>
> Our method was tested on DealOrNoDeal, a long-horizon task with a maximum length of *100 timesteps*, showing that the method can work with longer horizons. For negotiation styles like “stubborn” and “froward” (see Appendix A.6), the maximum horizon length was reached 2 and 22 times respectively during training. **The LLM correctly labeled 100% and 95% of those, respectively, indicating that the LLM is able to give accurate reward signals for long horizons.** We do not include an example of a long horizon negotiation in Fig. 1 due to space. We also clarified that the DealOrNoDeal task has a maximum horizon length of 100 in the paper (Sec. 4.3). For even longer horizons (>100), we strongly believe LLMs should be able to do well given their ability to perform on many other long-horizon tasks such as text-summarization tasks [3].
>
> > Q3. “ Can authors explain why sparse reward provided by your approach actually work? What's the difference if we use a sparse intermediate reward function for training? Will that actually work?”
>
> Although our method can be easily extended to provide non-sparse rewards, we believe that one reason why sparse rewards work for DealOrNoDeal is that agents are trained on a dataset of human negotiations using supervised learning before being fine-tuned with RL (we added training details in Appendix A.4). Thus, agents do not need to learn how to negotiate from scratch during RL. We adopt this method of training (fine-tuning supervised learning agents with sparse-reward RL) from prior works in the DealOrNoDeal domain [4,5]. Our other two domains are a single-timestep, and thus not sparse.
>
> **Our method can easily be used with non-sparse rewards by querying the LLM more frequently during an episode, instead of at the end of an episode. **
>
> * [1] Leibo et al., (2021). Scalable Evaluation of Multi-Agent Reinforcement Learning with Melting Pot
> * [2] Fan et al., (2022). MineDojo: Building Open-Ended Embodied Agents with Internet-Scale Knowledge
> * [3] Sanh, Webson, Raffel, Bach et al., (2022) Multitask Prompted Training Enables Zero-shot Task Generalization
> * [4] Lewis et al., (2017) Deal Or No Deal? End-to-End Learning for Negotiation Dialogues
> * [5] Kwon et al., (2021) Targeted Data Acquisition for Evolving Negotiation Agents

---

> ### Author Response · Authors · 2022-11-12
> **We Include a Pilot User Study where Users Find Our Agents to Be Well-Aligned.**
>
> > Q4. “ I feel it should be necessary to conduct some user study with human evaluation”
>
> During the rebuttal phase, we conducted a within-subjects pilot user study (N=10) in the DealOrNoDeal task. **We focus this study to determine whether our trained agents can meaningfully align themselves with different user objectives when the objectives are hard to define and users rate agent performance. Users find our agents to be significantly well-aligned compared to agents trained with a different objective, p < 0.001.** We have added results to Section 4.3.2 of the paper.
>
> We asked 10 users to pick a style in which they wanted their agent to negotiate in. Then we ask the users to provide positive and negative examples for their style by selecting example negotiations from a library of different negotiation behaviors. From this library, the user selects 3 negotiation behaviors, where Alice does or does not demonstrate their style (with short explanations). **Importantly, we had no ground truth access to what these styles should be defined as, making it difficult to engineer any type of reward.** We then trained a negotiation agent using the user-provided examples and explanations.
>
> To illustrate how effectively our framework can produce aligned agents, we also trained an agent in the opposing style by flipping the “Yes/No” labels of the user-provided explanations. We should expect users to perceive a large difference between the two agents.
>
> We asked each user to rate (from 1 - least aligned to 5 - most aligned) how aligned both negotiation agents are to their preferred style on 10 test negotiations.  Agents trained with the correct style were significantly more aligned (avg. $3.72 \pm 1.2$) than agents trained with the opposite style (avg. $1.56 \pm 1.05$), $p < 0.001$. This result suggests that our framework is truly versatile and is able to produce agents aligned with differently specified objectives by changing the explanations in the prompt. More details on the methods of this study can be found in Sec. 4.3.2.

---

### Author Response · Authors · 2022-11-12
**General Response**

We thank all the reviewers for their thorough reviews. The reviewers thought our work is a *“novel and promising formulation of using LLMs to specify rewards”*. Our work was *“well-executed”* with *“good experimental design”* on a *“simple (being good) idea”*, and our paper is *“well-written”* and *“easy-to-read”*.

Summary of common concerns:
* (1) Reviewers 7igZ, 1rSQ suggest the need for an experiment where the reward is hard to describe, and where we evaluate our framework with humans.
* (2) Reviewers UKq1, 49Yr would like us to explain our contribution regarding additional related works on using language for reward design/RL training.
* (3) Reviewers 49Yr, 1rSQ have concerns about generalization: generalization to longer horizon tasks, generalization to smaller LLM sizes.

We addressed all the concerns in the individual comments. We list a summary of changes we made to the paper here (major changes colored blue in the paper):
* (1) Conduct a pilot study with humans on tasks where reward is hard to describe and show that our framework can produce objective-aligned agents as evaluated by users.
* (2) Add relevant literature and highlight the novelty of our work
* (3) Conduct new experiments to show generalizability of method: evaluate with smaller LLM sizes.
* (4) Reproducibility: we add a Reproducibility Statement and include the code to generate prompts in the Supplementary. We will release all code with the camera ready version of our paper. We did not include the rest of the code due to size constraints.
* (5) Conduct new experiments to clarify experiment details (e.g., regarding the importance of $\rho_1$ and the importance of explanations).

We hope we have properly addressed the reviewer's concerns, and we would be happy to answer any further questions or concerns that the reviewers might have during this discussion period.

---

### Decision · Program_Chairs · 2023-01-20

**Decision:**

Accept: poster

**Justification For Why Not Higher Score:**

Given the remaining concerns of some reviewers and the AC, as listed in the meta review, it is more appropriate to accept the paper as a Poster.

**Justification For Why Not Lower Score:**

Generally speaking, the reviewers consider the proposed approach to be simple, intuitive, straightforward, and well-motivated, and the presents experimental results on controlling RL agents with LLM to be promising.

**Metareview: Summary, Strengths And Weaknesses:**

The paper proposes to use a large language model (LLM) to design rewards to train reinforcement learning (RL) agents. It converts both the desired behavior and the agent's behavior into textual prompts, which are evaluated under the LLM to output a reward signal. The reward signal is then used to optimize the RL agent's behavior. The proposed approach can work well even only given a single or a few demonstrations of the desired behaviors. Experiments in several different tasks, including long-horizon ones, show that the proposed approach can train RL agents aligned with the user's objectives and outperforms RL agents trained with supervisedly-learned reward functions.

Generally speaking, the reviewers consider the proposed approach to be simple, intuitive, straightforward, and well-motivated, and the presents experimental results on controlling RL agents with LLM to be promising.

There are, however, some disagreements on novelty, clarity, practicality, and performance limitation.

1) Novelty: Reviewer UKq1 does not consider the proposed approach to be that novel, as "somehow unsurprisingly given recent literature, leveraging these models to drive agents with textual prompts is possible in few-shot or zero-shot settings," and he/she feels that "its findings are not sufficient for the level of ICLR."

2). Clarity: the paper could be difficult to read for people who are not familiar with the considered tasks and how to convert behaviors into textual prompts.

3): Practicality: the proposed approach might not generalize well into the settings where behaviors cannot be easily concerted into textual prompts; it could be very difficult to appropriately design the g and f required in the proposed algorithm.

4) Performance limitation: the zero-shot and few-short ability of an LLM is a limiting factor of the performance of the RL agent; if considering LLM as the teacher, the best the RL agent can do is probably performing as good as the teacher; it is unclear whether the RL agent can outperform the teacher by combing the LLM produced reward and its own policy learning.




**Note From Pc:**

if the above contains the word "oral" or "spotlight" please see: "oral" presentation means -> notable-top-5% and "spotlight" means -> notable-top-25%. As stated in our emails, we are disassociating presentation type from AC recommendations

**Summary Of Ac-Reviewer Meeting:**

The AC was not able to find a meeting time that works for all by the deadline. The AC was able to separately meet both a reviewer who gave 8 and a reviewer who gave 5. The reviewer who gave 8 maintained his strong support for the paper, given its simplicity and effectiveness. The reviewer who gave 5 acknowledged that most of his concerns had been answered, and he would be considering responding to the authors and revising his/her rating. In addition, another reviewer who gave 5 responded in email that while he/she was not fully convinced about the contribution of the paper, he/she won't oppose if other reviewers think the paper can be useful.